# Public Perceptions on the Policy of Electronic Cigarettes as Medical Products on Twitter

**DOI:** 10.3390/ijerph20032618

**Published:** 2023-02-01

**Authors:** Xubin Lou, Pinxin Liu, Zidian Xie, Dongmei Li

**Affiliations:** 1Goergen Institute for Data Science, University of Rochester, Rochester, NY 14627, USA; 2Department of Computer Science, University of Rochester, Rochester, NY 14627, USA; 3Department of Clinical and Translational Research, University of Rochester Medical Center, Rochester, NY 14642, USA

**Keywords:** electronic cigarettes, medical products, Twitter, public perception

## Abstract

Starting from 1 October 2021, Australia requires a prescription for purchasing nicotine vaping products. On 29 October 2021, the UK provided a guideline to treat e-cigarettes as medical products. This study aims to understand public perceptions of the prescription policy in Australia and the UK on Twitter. Tweets related to e-cigarettes from 20 September 2021 to 31 December 2021 were collected through Twitter streaming API. We adopted both a human and machine learning model to identify a total of 1795 tweets from the UK and Australia related to the prescription policy. We classified them into pro-policy, anti-policy, and neutral-to-policy groups, and further characterized tweets into different topics. Compared to Australia, the proportion of pro-policy tweets in the UK was significantly higher (19.43% vs. 10.92%, *p* < 0.001), while the proportion of anti-policy tweets was significantly lower (43.4% vs. 50.09%, *p* = 0.003). The main topics for different attitudes towards the prescription policy between the two countries showed some significant differences, for example, “help quit smoking” in the UK and “health effect of e-cigarettes” in Australia for the positive attitude, “economic effect” in the UK and “preventing smoking cessation” in Australia for the negative attitude, which reflected different public concerns. The findings might provide valuable guidance for other countries to implement a similar policy in the future.

## 1. Introduction

Among youth, e-cigarettes, especially disposable e-cigarettes, are more popular than any traditional tobacco product [1]. According to the 2021 National Youth Tobacco Survey, more than 2 million U.S. middle and high school students reported e-cigarettes use in 2021, with more than 8 in 10 of those youth using flavored e-cigarettes [2]. In the United Kingdom (UK), the proportion of current vapers who have vaped for more than 3 years appears to be increasing (23.7% in 2018, 29.3% in 2019, 39.2% in 2020) [3]. The proportion of young people trying e-cigarettes first has increased to 24.6% in 2021 [4]. In Australia, e-cigarette use by Australians aged 14 or older has more than doubled from 2016 to 2019 [5]. Recent studies on e-cigarettes have indicated the potential health risks of e-cigarettes on the lungs, cardiovascular system, and mental health [6,7,8]. There has been an outbreak of lung injuries and deaths associated with vaping. In February 2020, the Center for Disease Control and Prevention (CDC) confirmed 2807 cases of e-cigarette or vaping use-associated lung injury (EVALI) and 68 deaths attributed to that condition [9].

Starting from 1 October 2021, all nicotine vaping products in Australia, such as nicotine e-cigarettes, nicotine pods, and liquid nicotine, are Schedule 4 (prescription only) medicines. The legislative changes in Australia mean that consumers require a valid prescription from their general practitioner (GP) for all purchases of nicotine vaping products [10]. Other countries, including the U.S., Canada, and the UK, also have prescription systems in place, but Australia is the only country in the world to restrict access to prescription-only. To reduce the smoking rate and eliminate the stark disparities in smoking rates, the National Health Service (NHS) in England also plans to soon require the prescription of e-cigarettes [11]. On 29 October 2021, a guidance for licensing electronic cigarettes and other inhaled nicotine-containing products as medicines in the UK was explicit on the official website [12].

As one of the most popular social media platforms, with 353.9 million users, and a prevalence of e-cigarettes-related posts, Twitter is an ideal avenue to investigate public perceptions and discussions of the recent legislative changes in Australia that treat e-cigarettes as prescription medicines [13]. Twitter data have been successfully used to understand public perceptions of several regulatory policies related to flavored e-cigarettes in the U.S. [14,15,16].

This study aims to analyze public perceptions and discussions of the e-cigarette prescription policy in Australia and the potential similar policy in the UK using Twitter data. We have compared the major differences in public perception and discussion of this policy on Twitter between the UK and Australia. Our results could help the policymakers understand the public attention and concerns regarding the prescription policy in the UK and Australia, which might offer valuable guidance and suggestions for other government agencies to make similar e-cigarette prescription policies in the future.

## 2. Materials and Methods

### 2.1. Data Collection and Preprocessing

Twitter posts (tweets) related to e-cigarettes from 20 September 2021 to 31 December 2021 were collected through the Twitter streaming API (Application Programming Interface) using keywords related to e-cigarettes or vaping, such as e-cigarette, ecig, and vaping [17]. First, we identified the related tweets by filtering with policy-related keywords, removing the commercial tweets, and identifying tweets from the UK or Australia. Secondly, we transformed the collected tweets corpus into a format with only its original context and deleted all tags and mentions. Thirdly, we adopted a human-guided machine-learning framework by manually labeling sampled tweets as the training data for machine learning to classify the tweets into pro-, anti-policy, and neutral categories (Appendix A).

To identify tweets related to the policy on e-cigarettes as medical products, further filtering was performed using a list of keywords, including “vaping”, “vape”, “e-cig”, “electroniccigarette”, “vapor”, “e-liquid”, “ejuice”, “eliquid”, “policy”, “vapin”, “tobacco”. We also included more than 30 ministers’ names and agencies’ names as search keywords while filtering the data, such as “alexwodak”, “genews”, and “sajidjavid”. We lowercased all search strings for regular expression matchings.

The following set of keywords were used to filter out unrelated commercial and promotion tweets from the dataset: “deal”, “supply”, “dealer”, “customer”, “discount”, “sale”, “free shipping”, “sell”, “$”, “%”, “dollar”, “offer”, “percent off”, “store”, “promo”, “promotion” [14]. The names of states and cities with the top 30 populations in the United Kingdom and Australia were used to identify the geographical location of the tweets. For example, “Scotland”, “Wales”, “Northern Ireland”, “London”, “Birmingham”, “Glasgow”, “Liverpool”, “Bristol”, “Manchester” in the UK, and “Queensland”, “Stray”, “Sydney”, “Brisbane”, “Melbourne”, “New South Wales”, “Springfield” in Australia. The full names of these locations and their abbreviations were used for filtering. Overall, we identified a dataset containing 9368 related tweets from Australia and the UK. We utilized a tweet preprocessing algorithm which uses a long short-term-memory (LSTM) model to encode sentences and automatically detect hashtags and quoted names within the tweets and transform these words into special tokens to ease human and machine annotation and further analysis [18]. This helps remove the irrelevant tags and mentions in the tweets that may interfere with further annotation.

To avoid possible noise from the general discussion on vaping and drug regulations, we implemented a random sampling based on the timeframe 20 days before or after the policy announcement date to select 2775 tweets from 9368 tweets containing keywords. By hand-coding, among the selected 2775 tweets, only 996 discussed e-cigarette prescription policy in either the UK or Australia. To strike a balance between automation and accuracy, we checked the left dataset through a state-of-the-art transformer language model XLM-Roberta to label the relevance of the other tweets from the whole dataset. XLM-Roberta is a deep learning model with a strong ability in cross-lingual natural language understanding and can be used to extract features useful for downstream tasks [19]. We added one layer of the feedforward network to the original model for the binary classification for relevance detection. To achieve this, we randomly selected 80% of relevant and irrelevant tweets from the hand-coded 2775 tweets and shuffled them into the training corpus to ensure the balance of the training corpus. The remaining tweets were treated as the validation corpus to test the model’s accuracy. This framework automatically searches for relevant tweets to increase the sizes of the relevant datasets. The model indicated a strong ability for relevant tweet detection with a final precision score of 0.853, a recall score of 0.903, and an F1 score of 0.877.

We identified 946 tweets from the rest of the 9368 tweets based on the machine-learning model. We manually checked these tweets and identified 799 relevant tweets, which further indicates the robustness of our model in relevance identification. Together, we have identified 1795 tweets related to the policy about e-cigarettes as medical products. Among them, 1090 tweets were from Australia, and 705 were from the UK.

### 2.2. Content Analysis

To develop the sentiment and topic codebook, we implemented a simple random sampling to select 300 posts for hand-coding from the final dataset. Two human coders independently coded the tweet context. Any discrepancy was resolved by the discussion within a group of four members. We reached a kappa statistic of 0.89, indicating a high agreement.

Tweets were classified into three main categories: pro-policy (e.g., supporting the prescription policy), anti-policy (e.g., disagreeing with the government control or emphasizing the side effect of the policy), and neutral to the policy (e.g., simply describing the policy or referring to the recent news without holding personal sentiment towards the policy).

Tweets with pro-policy were categorized as (1) health effect of e-cigarettes, for example, tweets describing the negative health effect of e-cigarettes on lungs, like lung cancer; (2) effect on youth, for example, tweets that emphasizes the current youth addiction of the e-cigarettes to support the policy announcement; (3) help quit smoking, for example, tweets saying the new policy will reduce the smoking rate; (4) professional guidance, for example, tweets describing that doctors will be the agent that helps provide prescriptions so that people can get professional suggestions and instructions on vaping; (5) other reasons, for example, tweets praising the policy without giving specific reasons or providing reasons other than the categories as mentioned above, belonging to a subtle group.

Topics for tweets with anti-policy included (1) not help with smoke cessation, for example, tweets warning that the policy will make vapes less accessible so that people will choose cigarettes instead; (2) hard to vapers, for example, tweets stating that long time vapers will no longer get the e-cigarette products; (3) GP’s misinformation, for example, tweets emphasizing that doctors did not get enough trainings or instructions on how to prescribe e-cigarettes; (4) emotional catharsis, for example, tweets showing a cynical attitude with rude words; (5) policy is not useful, for example, tweets considering the policy will have little effect or is unnecessary; (6) economic effect, for example, tweets disapproving the policy because of the complaint of taxpayer funding used to promote the policy, more urgent funding needs in improving other services, unnecessary promotion of e-cigarette use with the already cheaper price of e-cigarettes in the market, and potential causation of black market for vaping products; (7) personal choice, for example, tweets proposing that whether to buy e-cigarettes should be considered a personal choice instead of regulated by an official agency; (8) other regulatory needs, for example, tweets claiming the regulation of other tobacco products is needed; (9) other reasons, tweets showing negative views towards the policy without giving appropriate reasons or providing reasons other than the above categories, belonging to a subtle group. 

### 2.3. Statistical Analysis

To investigate whether Twitter users from the United Kingdom (UK) and Australia show similar attitudes towards the e-cigarette prescription policy, we performed a Chi-square test on the general pro-policy, anti-policy, and neutral attitudes to see whether the overall distribution of the attitude was significantly different between the two countries. In addition, we conducted two-proportion z-tests in each attitude category to examine the differences between the UK and Australia. We conducted a two-proportion z-test to compare the proportion of each topic within either anti-policy or pro-policy between the UK and Australia.

## 3. Results

### 3.1. Longitudinal Trend in the Mentions of the e-Cigarette Prescription Policy on Twitter in the UK and Australia

To capture all tweets related to the e-cigarette prescription policy, including before and after the announcement, we collected the relevant tweets from 20 September 2021 to 31 December 2021. As shown in Figure 1, the discussion about the e-cigarette prescription policy in Australia starts with a nearly exponential increasing trend from 25 September 2021, until it reaches a peak (130 tweets) on 1 October 2021. Then, it follows a gradual decrease from the peak to around 27 October 2021, with a few short-term fluctuations in between. On the other side, in the UK, there was an extreme increase in the policy discussion on 29 October 2021, but it suddenly dropped within 5 days. 

### 3.2. Attitudes towards the e-Cigarette Prescription Policy on Twitter in the UK and Australia

As shown in Figure 2, among relevant tweets in the UK, 137 (19.43%, 137/705) are pro-policy, 306 (43.40%, 306/705) are anti-policy, and 262 (37.16%, 262/705) are neutral. In contrast, in Australia, 119 (10.92%, 119/1090) are pro-policy, 546 (50.09%, 546/1090) are anti-policy, and 425 (38.99%, 425/1090) are neutral. Results from the Chi-square test showed a significantly higher proportion of positive attitudes toward the prescription policy in the UK than that in Australia (19.43% vs. 10.92%, *p* < 0.001). While the negative attitudes towards the policy were dominant in both countries, Australia showed a significantly higher proportion of negative attitudes toward the prescription policy than the UK (50.09% vs. 43.4%, *p* = 0.003). We did not observe a significant difference in the proportion of neutral attitudes between the UK and Australia (37.16% vs. 38.99%, *p* = 0.233).

### 3.3. Topics in Tweets Mentioning the e-Cigarette Prescription Policy in the UK and Australia

To investigate what might lead to different perceptions towards the e-cigarette prescription policy between the UK and Australia, we compared the topics in tweets with either pro- or anti-policy attitudes between the two countries (Appendix A). As shown in Figure 3, the main differences in positive attitudes towards the policy between the UK and Australia were: help quit smoking, health effect of e-cigarettes, and effect on youth. The dominant pro-policy topic (65.69%, 90/137) in the UK considered that the prescription of e-cigarettes could help people quit smoking. In contrast, it was only 15.13% in Australia, which was significantly lower than that in the UK (*p* < 0.001). On the other hand, there were more discussions that the policy was more likely to reduce e-cigarette’s negative health effects in Australia than those in the UK (47.90% vs. 16.06%, *p* < 0.001). In addition, 21% of pro-policy tweets from Australia were about that the policy had a positive effect on relieving youth addiction, while it was only 6.57% in the UK (*p* < 0.001). There was no significant difference in professional instruction and others between the two countries.

As shown in Figure 4, the dominant anti-policy topic in the UK was concerns about the negative economic impacts of policy (56.86%, 174/306). In contrast, it was only 13.19% in Australia, which is significantly lower than that in the UK (*p* < 0.001). A higher proportion of anti-policy tweets in Australia claimed that limited access to vaping products might not help with smoking cessation than in the UK (26.92% vs. 4.58%, *p* < 0.001). Other topics showing a significant difference between Australia and UK included Other regulatory needs (12.27% vs. 1.63%, *p* < 0.001), GPs’ misinformation (10.26% vs. 4.25%, *p* < 0.001), hard to vapers (10.07% vs. 1.63%, *p* < 0.001), and personal choice (2.56% vs. 5.88%, *p* = 0.012).

## 4. Discussion

In this study, we investigated the difference in public perceptions of the e-cigarette prescription policy on Twitter between Australia and the UK from 20 September 2021 to 31 December 2021. Both countries had an explicit dominant negative attitude towards this policy. Tweets in the UK were more likely to have positive attitudes toward the policy than those in Australia. In contrast, there were more tweets with negative sentiments toward the policy in Australia than in the UK. The major difference in the pro-policy view was that most tweets in the UK considered the policy helping people quit smoking with more available prescribed e-cigarettes. In Australia, Twitter users were more likely to think that the policy could reduce the negative effect of e-cigarette products on health by limiting the consumption of vaping in the prescription policy. For anti-policy tweets, the most popular topic in the UK was the economic impact, while in Australia, Twitter users were worried that the restriction on accessing e-cigarettes makes it harder for people to quit smoking.

Along with the previous studies, most of the relevant tweets we collected appeared within one week when the governments made the announcements [20,21]. The peak for the number of policy-related tweets identified on Twitter in Australia corresponds to the official release date of the medical prescription policy in Australia, and then followed a volatile downtrend [22]. Similarly, on 29 October 2021, we observed an apparent peak in the UK with a dominant anti-policy view of economic effect when the UK Medicines and Healthcare Products Regulatory Agency proposed guidance for prescribing electronic cigarettes as medical products to the National Health Service (NHS) [12]. Overall, Australia seems to have a more long-lasting effect. On the other side, the discussion in the UK was more transient and dramatic. The reason for this difference could be that the policy in Australia has already been implemented, while the policy in the UK currently is just a proposal or guidance in the discussion about the potential future policy.

The dominant pro-policy opinion in the UK considered that the prescription of e-cigarettes could help people quit smoking. Most tweets sharing this view believed e-cigarettes would be accessible through prescription, which can help replace the traditional nicotine products and help smokers quit smoking. On the other hand, Twitter users in Australia tended to believe that the policy was more likely to resolve e-cigarette’s health effects and prevent youth addiction. With a 96% increase in e-cigarette usage among young people, the Australian government aimed to reduce youth addiction and treated e-cigarettes as poison and hazardous substances to be regulated [23,24,25]. Nevertheless, the guidelines from the NHS in the UK intended to relieve or prevent craving and nicotine withdrawal symptoms when tobacco smokers wish to quit or reduce smoking [12].

The most popular anti-policy topic in the UK is the concern of the negative economic impact. Compared with Australia, Twitter users in the UK were much more likely to think e-cigarettes are cheaper than conventional tobacco products so the government should not provide e-cigarettes as subsidized medical prescriptions for smokers as they can afford. This dominant effect comes with the innate policy change in e-cigarette regulation. Originally regulated as customer products, e-cigarettes are subject to a 20% value-added tax (VAT). However, treated as a medical product, e-cigarettes are subject to a reduced VAT of 5% [26,27]. This subsidy with an extensive regulatory cost during both pre- and post-authorization, further increased the financial burden to the NHS [28]. Therefore, many discussions around the unwillingness to pay the bills appeared in our Twitter dataset, such as “E-cigs on the NHS? if smokers want to quit, they’ll quit, vapes aren’t that expensive, much cheaper than cigarettes. Instead of spending millions on that, use it for …”. The majority (125/174, 71.84%) of tweets in the UK about negative economic effects are explicit complaints of huge taxpayer funds spent in pushing the policy, which is an unnecessary investment and unfair to non-smokers or non-vapers. The cost incentive to switch is clear, so a taxpayer subsidy is not required. However, the economic considerations in Australia are more likely to raise a warning that the policy would bring a black market for vaping products [29]. Our result agrees with previous findings that the lengthy pharmaceutical approval process and extensive regulatory burden could result in black markets for unregulated e-cigarettes purchase [30]. A large proportion (40/72, 55.56%) of tweets in Australia about negative economic effects mentioned the concern of the black market and potential corruption between the government and vape manufacturers. Compared with the healthcare system in the UK, the Australian system is more decentralized than the British one, which makes it more pluralistic. Most Australians have private health insurance (PHI), which makes Australians less dependent on the public healthcare system less and can receive a rebate in the Australian tax and transfer system. In consequence, Australia achieves substantially better healthcare outcomes than the UK and total healthcare spending is lower in Australia. Better private health insurance implementation and relatively less reliance on the public healthcare system might be one of the potential reasons why Australians are less sensitive to the policy tax than the British [31].

The two most popular topics in anti-policy tweets in Australia included that the e-cigarette prescription policy would not help with smoking cessation and it would be hard for long-time vapers while tweets in the UK did not present much concern. The difference in the prescription policy between the two countries is likely to reflect disparate attitudes. Instead of supporting e-cigarettes as medical products with subsidies as in the UK, Australia’s policy restricted both the importation and access of e-cigarettes from the market by requiring the practitioner’s permission and prescription [22]. Similarly, under the e-cigarette restrictions, Twitter users in Australia are more likely to argue that other products, like conventional cigarettes, should be regulated if the government aims to reduce the smoking rate. Gibson’s study shows that UK smokers report fewer quit attempts than those in Australia, Canada, and the U.S., but are more likely to use support when quitting and to achieve short-term abstinence than smokers in the other countries, which particularly provides the underlying reason for UK’s less concern about the e-cigarette prescription in preventing smoking cessations since they were shown to be more likely to quit smoking [32]. Regarding the public concern of improper policy regulations in the two countries, Twitter users in Australia are more likely to complain that only a few general practitioners were familiar with the e-cigarette prescription regulation, and many doctors could not provide professional instructions. For example, “Many Doctors have been exposed to poor guidance from @RACGP &amp.” Currently, available literature has revealed GP’s knowledge deficit around e-cigarettes for smoking cessation [33,34,35]. The statistics in our study based on tweets in Australia reflected this notion. However, we did not find a similar pattern in the UK, which could be because the UK policy has not been officially implemented.

Our study has several limitations. First, we only analyzed Twitter data in this study. However, Twitter users are not representative of the whole population. Secondly, the keywords we used are not complete, leading to some missing tweets in the corpus. Thirdly, during the data preprocessing section, the performance of the XLM-Roberta model of this study showed a precision score of 0.85, which indicated we might miss some truly relevant tweets in our data analyses. Thus, the dataset may not be completely representative of the overall distribution of the sentiments towards the policies in the two countries. Fourthly, there was only limited demographic information available for some Twitter users who posted relevant tweets. We could not distinguish the attitudes from different demographic groups and determine who were vapers among these Twitter users, thus losing some potential reflections on the distribution of attitudes towards the policy with different user demographic groups.

## 5. Conclusions

Twitter users in Australia were inclined to show a more negative sentiment towards the e-cigarette prescription policy than Twitter users in the UK. Australia’s pro-policy tweets focused on health effects of e-cigarettes and youth addiction, while the UK’s dominant pro-policy argument was the policy’s effectiveness in helping reduce smoke rates. Twitter users in Australia were more likely to have concerns that the policy would prevent smoking cessation, while Twitter users in the UK were more pessimistic about the economic effect. Our findings about public perception of this policy on Twitter could help public health authorities better understand public concerns, which will be helpful for future policymaking, especially in the UK or other countries where this policy has not been implemented yet.

## Figures and Tables

**Figure 1 ijerph-20-02618-f001:**
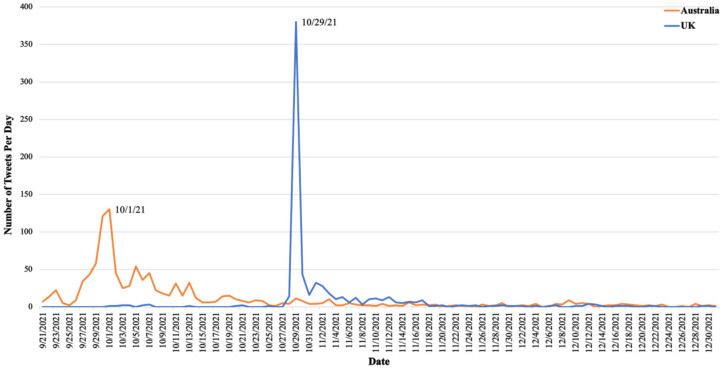
Longitudinal trends of tweets mentioning e-cigarettes as medical products in the UK and Australia.

**Figure 2 ijerph-20-02618-f002:**
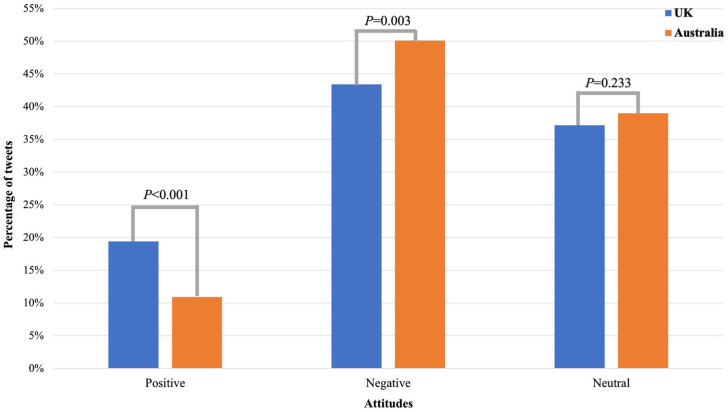
Attitudes towards the e-cigarette prescription policy on Twitter in the UK and Australia.

**Figure 3 ijerph-20-02618-f003:**
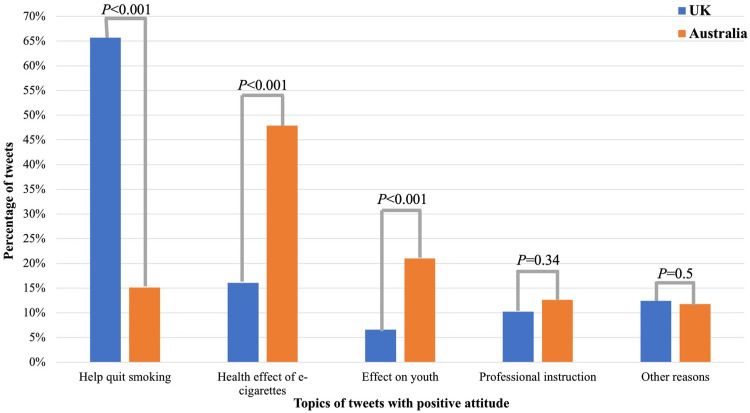
Topics in tweets with positive attitudes towards the e-cigarette prescription policy in the UK and Australia.

**Figure 4 ijerph-20-02618-f004:**
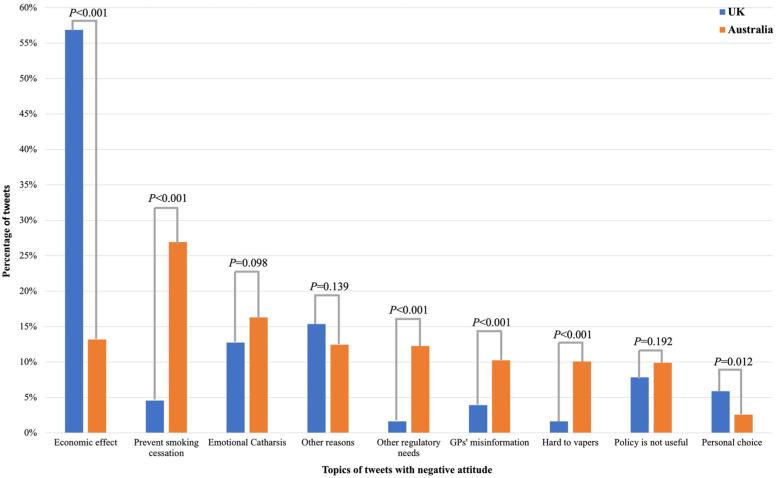
Topics in tweets with negative attitudes towards the e-cigarette prescription policy in the UK and Australia.

## Data Availability

The social media data and Python code used for data analysis will be available upon reasonable request from the corresponding authors.

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
