# Peer review of "Public Perceptions on the Policy of Electronic Cigarettes as Medical Products on Twitter"

_ijerph, 2023, doi:10.3390/ijerph20032618_

Round 1

Reviewer 1 Report

This is an interesting article addressing a pertinent public health issue of the modern world.
The manuscript is drafted in a good fashion.
Some areas of concern are as follows

There should be a clear-cut explanation on the fact why the data collection period was chosen to be from September to December 2021 when the actual policy/guideline was announced on 1st and 29th Oct 2021.
Line 53; 329 million twitter users was mentioned, however currently the data suggests different figure, thus it needs to be corrected as well as appropriately cited

Line 61; We will compare....., better should write we have compared in this manuscript.....

Lines 92-4; "tweet processing algorithm" needs to be elaborated in a sentence, for the ease of readers.

Line 301; "there was no demographic information about Twitter users who posted relevant 301 tweets", this sweeping statement seems to be false as we can find some sort of demographic information for various twitter users in their respective profile information. However, this lack of demographic information is a serious limitation in this study, as the study population has its unique importance in overall impact of their opinions in such studies.

Author Response

Comments and Suggestions for Authors

This is an interesting article addressing a pertinent public health issue of the modern world. The manuscript is drafted in a good fashion.

Response: Thanks for the encouraging comments.

Some areas of concern are as follows

There should be a clear-cut explanation on the fact why the data collection period was chosen to be from September to December 2021 when the actual policy/guideline was announced on 1st and 29th Oct 2021.

Response: Thank you for your feedback. While most of discussion on social media occurs on the day when the e-cigarette prescription policy was announced, some relevant discussion can happen prior the announcement and last a long time after the announcement. Therefore, in order to capture all relevant tweets, we purposely extended our data collection period. Our results in Figure 1 did support this. We have added this at the beginning of the Results section “To capture all tweets related to the e-cigarette prescription policy, including before and after the announcement, we collected the relevant tweets from September 20, 2021, to December 31, 2021”.

Line 53; 329 million twitter users was mentioned, however currently the data suggests different figure, thus it needs to be corrected as well as appropriately cited

Response: We have updated this number from 329 million to 353.9 million.

Line 61; We will compare....., better should write we have compared in this manuscript.....

Response: We have made the changes as suggested.

Lines 92-4; "tweet processing algorithm" needs to be elaborated in a sentence, for the ease of readers.

Response: As suggested, we have added “We utilized a tweet preprocessing algorithm which uses long short-term-memory (LSTM) model to encode sentences and automatically detect hashtags and quoted names within the tweets and transform these words into special tokens to ease human and machine annotation and further analysis.”

Line 301; "there was no demographic information about Twitter users who posted relevant 301 tweets", this sweeping statement seems to be false as we can find some sort of demographic information for various twitter users in their respective profile information. However, this lack of demographic information is a serious limitation in this study, as the study population has its unique importance in overall impact of their opinions in such studies.

Response: We agree that the demographic information can be very valuable in this study. We understand that for some Twitter users, some sort of demographic information can be available. Yet, they are very limited depending on individual Twitter user. In addition, some demographic information (such as age and gender) could be estimated by analyzing the user profile image with face recognition algorithms. However, this information is limited since the estimation is not accurate and more importantly most Twitter users did not share their selfies.  Together, it might be more reasonable for us to put this as one of limitations “there was only limited demographic information available for some Twitter users who posted relevant tweets”.

Reviewer 2 Report

Minor corrections:

L232: Should 'align' be Along?

L262: financial burden 'to' the NHS

Author Response

Minor corrections:

L232: Should 'align' be Along?

Response: Thank you for your feedback. We have made the changes as suggested.

L262: financial burden 'to' the NHS

Response:  Thank you for your feedback. We have made the changes as suggested.

Reviewer 3 Report

This is a generally sound, well written descriptive paper on the topic, which should be of considerable interest to readers, given the importance of Twitter as a communication mechanism.  The limits of the study as well as the methods are extensively discussed.  My major suggestion for improvement is that there might be some discussion at the end about possible underlying reasons for the differences in findings. I realize this is hampered by the lack of relevant demographic data in Twitter, but some discussion about cultural attitudes toward public health in the two countries and how these relate to the findings here would be valuable, at least tentatively.

The covid-19 pandemic showed that there are different degrees of receptivity among countries and groups to public health messages.  In the paper at hand, are there documented, discernible differences in receptivity to anti-smoking messages between Australia and England that might help account for the patterns found in the data?  This is what political scientists and journalists call political culture.  See, for instance, jacob sullum, for your own good: the anti-smoking crusade and the tyranny of public health (1998)

Author Response

This is a generally sound, well written descriptive paper on the topic, which should be of considerable interest to readers, given the importance of Twitter as a communication mechanism.  The limits of the study as well as the methods are extensively discussed.  My major suggestion for improvement is that there might be some discussion at the end about possible underlying reasons for the differences in findings. I realize this is hampered by the lack of relevant demographic data in Twitter, but some discussion about cultural attitudes toward public health in the two countries and how these relate to the findings here would be valuable, at least tentatively.

The covid-19 pandemic showed that there are different degrees of receptivity among countries and groups to public health messages.  In the paper at hand, are there documented, discernible differences in receptivity to anti-smoking messages between Australia and England that might help account for the patterns found in the data?  This is what political scientists and journalists call political culture.  See, for instance, jacob sullum, for your own good: the anti-smoking crusade and the tyranny of public health (1998)

Response: Thanks for your very helpful feedback. In the Discussion section, we have added “Compared with the healthcare system in the UK, the Australian system is more decentralized, which makes it more pluralistic. Most Australians have private health insurance (PHI), which makes Australians less dependent on the public healthcare system and can receive a rebate in the Australian tax and transfer system. In consequence, Australia achieves substantially better healthcare outcomes than the UK and total healthcare spending is lower in Australia. Better private health insurance implementation and relatively less reliance on the public healthcare system might be one of the potential reasons why Australians are less sensitive to the policy tax than the British” and “Gibson’s study shows that U.K. smokers report fewer quit attempts than those in Australia, Canada, and the US, but are more likely to use support when quitting and to achieve short-term abstinence than smokers in the other countries, which partially provides the underlying reason for UK’s less concern about the e-cigarette prescription in preventing smoking cessations since they were shown to be more likely to quit smoking”.

Reviewer 4 Report

This paper aims to compare public perceptions and discussions of an e-cigarette prescription policy in Australia and guidelines for a potential similar policy in the UK using Twitter data. The results may help policymakers understand public attention and concern regarding e-cigarette prescription policies currently and in future. The main strength of this paper is its relevance to policy approaches that aim to address or regulate the use of e-cigarettes containing nicotine. Other strengths are its novel methods of mining twitter discussion using both human and machine learning.

This paper contributes to a gap in understanding public responses to policy approaches regarding the regulation of e-cigarettes. The use of a human and machine learning model is both novel and appropriate for the type of data being analyzed, despite the limitations outlined in the discussion. This paper should be of interest to policymakers and researchers from a policy and methodological perspective. 

Other than a few minor inconsistencies identified in my specific comments below, this paper is well structured, clear and relevant.

Abstract: line 21-24 Re. "The main topics for different attitudes towards the prescription policy between the two countries showed some significant differences. Our results showed that public perceptions of the prescription policy on e-cigarettes on Twitter between the UK and Australia were different, which resulted from different public concerns."

-These two sentences basically state the same thing and are not very helpful to the reader. I suggest reducing the duplication and briefly including what some of the main topics of differences in attitude were (at a high level).

Introduction:

- Line 37: Data is presented on the increased prevalence of e-cigarette use in youth and young adults in the US, UK and Australia. I think these data are useful to show the trend and rationale for policy to regulate e-cigarettes. However, I found the data on e-cigarette use among young adults in Australia who are smokers to be confusing. This made me wonder whether the previous data presented was for all youth, non-smoking youth, or smoking youth. I think these data need further context.

- Line 49: "To help stop smoking tobacco products the National Health Service (NHS) in England also plans to soon require the prescription of e-cigarettes..." This sentence is a bit awkward. I suggest explaining how this fits into the UKs harm reduction approach to help people quit smoking tobacco products.

Materials and Methods:

-Line 90: Brisbane is repeated 

-Line 130: Awkward sentence. I'm not sure what is meant by 'cause lung cancer'?

Results:

-Line 195 and 199/200: Instead of 'Nicotine's health effect' do you mean 'E-cigarette's health effect'? In the content analysis section, the health effect of e-cigarettes is referred to as a category, not nicotine. This is a very important distinction as nicotine is just one component of an e-cigarette and associated harms. Table S1 and Figure 3, also use 'health effect of e-cigarettes'. This inconsistency also appears in the Discussion section (line 248) and Conclusions (line 309). 

Discussion

-Line 248: re. health effects of e-cigarettes vs. nicotine as pointed out in above comments

-Line 280: "Instead of supporting e-cigarettes as medical products with subsidies in the UK,..." I think '...as in the UK' is needed? Otherwise the sentence could be misinterpreted as having the opposite meaning.

Conclusions:

-Line 309: re. ''nicotine's health effects" should be health effects of e-cigarettes.

Author Response

This paper aims to compare public perceptions and discussions of an e-cigarette prescription policy in Australia and guidelines for a potential similar policy in the UK using Twitter data. The results may help policymakers understand public attention and concern regarding e-cigarette prescription policies currently and in future. The main strength of this paper is its relevance to policy approaches that aim to address or regulate the use of e-cigarettes containing nicotine. Other strengths are its novel methods of mining twitter discussion using both human and machine learning.

This paper contributes to a gap in understanding public responses to policy approaches regarding the regulation of e-cigarettes. The use of a human and machine learning model is both novel and appropriate for the type of data being analyzed, despite the limitations outlined in the discussion. This paper should be of interest to policymakers and researchers from a policy and methodological perspective.

Other than a few minor inconsistencies identified in my specific comments below, this paper is well structured, clear and relevant.

Response: Thanks for your positive comments.

Abstract: line 21-24 Re. "The main topics for different attitudes towards the prescription policy between the two countries showed some significant differences. Our results showed that public perceptions of the prescription policy on e-cigarettes on Twitter between the UK and Australia were different, which resulted from different public concerns."

-These two sentences basically state the same thing and are not very helpful to the reader. I suggest reducing the duplication and briefly including what some of the main topics of differences in attitude were (at a high level).

Response: Thank you for your feedback. We have removed the duplicates and added some major topics of differences as suggested “The main topics for different attitudes towards the prescription policy between the two countries showed some significant differences, for example Help quit smoking in the UK and Health effect of e-cigarettes in Australia for the positive attitude, Economic effect in the UK and Prevent smoking cessation in Australia for the negative attitude.”

Introduction:

- Line 37: Data is presented on the increased prevalence of e-cigarette use in youth and young adults in the US, UK and Australia. I think these data are useful to show the trend and rationale for policy to regulate e-cigarettes. However, I found the data on e-cigarette use among young adults in Australia who are smokers to be confusing. This made me wonder whether the previous data presented was for all youth, non-smoking youth, or smoking youth. I think these data need further context.

Response: Thanks for the comment. “e-cigarette use by Australians aged 14 or older has more than doubled from 2016 to 2019” was for all youth regardless of their smoking statuses. To avoid the confusion, we changed this sentence to “In Australia, e-cigarette use by Australians aged 14 or older has more than doubled from 2016 to 2019.”

- Line 49: "To help stop smoking tobacco products the National Health Service (NHS) in England also plans to soon require the prescription of e-cigarettes..." This sentence is a bit awkward. I suggest explaining how this fits into the UKs harm reduction approach to help people quit smoking tobacco products.

Response: By prescribing e-cigarettes as medical products, it becomes more accessible via the NHS for anybody who want to quit smoking tobacco products, which could help to narrow the stark disparities in smoking rates between rich and poor areas in England. To reflect this, we revised this sentence into “To reduce the smoking rates and eliminate the stark disparities in smoking rate, the National Health Service (NHS) in England …”.

Materials and Methods:

-Line 90: Brisbane is repeated

Response: We have removed the duplicate.

-Line 130: Awkward sentence. I'm not sure what is meant by 'cause lung cancer'?

Response: We have changed “cause lung cancer” to “like lung cancer”.

Results:

-Line 195 and 199/200: Instead of 'Nicotine's health effect' do you mean 'E-cigarette's health effect'? In the content analysis section, the health effect of e-cigarettes is referred to as a category, not nicotine. This is a very important distinction as nicotine is just one component of an e-cigarette and associated harms. Table S1 and Figure 3, also use 'health effect of e-cigarettes'. This inconsistency also appears in the Discussion section (line 248) and Conclusions (line 309).

Response: We have made them consistent with the “health effect of e-cigarettes”.

Discussion

-Line 248: re. health effects of e-cigarettes vs. nicotine as pointed out in above comments

Response: We have made them consistent with the “health effect of e-cigarettes”.

-Line 280: "Instead of supporting e-cigarettes as medical products with subsidies in the UK,..." I think '...as in the UK' is needed? Otherwise the sentence could be misinterpreted as having the opposite meaning.

Response: Thank you for your feedback. We have made the change as suggested.

Conclusions:

-Line 309: re. ''nicotine's health effects" should be health effects of e-cigarettes.

Response:  Thank you for your feedback. We have made the change as suggested.